# Improved Multi-Model Classification Technique for Sound Event Detection in Urban Environments

Muhammad Salman Khan [1], Mohsin Shah [2,*], Asfandyar Khan [3], Amjad Aldweesh [4], Mushtaq Ali [3], Elsayed Tag Eldin [5], Waqar Ishaq [2] and Lal Hussain [6,7,*]

1  Institute of Space and Technology, Islamabad 44000, Pakistan
2  Department of Telecommunication, Hazara University Mansehra, Mansehra 21120, Pakistan
3  Department of Computer Science & Information Technology, Hazara University Mansehra, Mansehra 21120, Pakistan
4  College of Computer Science and Information Technology, Shaqra University, Shaqra 11961, Saudi Arabia
5  Faculty of Engineering and Technology, Future University in Egypt, New Cairo 11835, Egypt
6  Department of Computer Science and Information Technology, Neelum Campus, University of Azad Jammu and Kashmir, Athmuqam 13230, Pakistan
7  Department of Computer Science and Information Technology, King Abdullah Campus, University of Azad Jammu and Kashmir, Muzaffarabad 13100, Pakistan
*  Correspondence: syedmohsinshah@hu.edu.pk (M.S.); lall_hussain2008@live.com (L.H.)

**Abstract:** Sound event detection (SED) plays an important role in understanding the sounds in different environments. Recent studies on standardized datasets have shown the growing interest of the scientific community in the SED problem, however, these did not pay sufficient attention to the detection of artificial and natural sound. In order to tackle this issue, the present article uses different features in combination for detection of machine-generated and natural sounds. In this article, we trained and compared a Stacked Convolutional Recurrent Neural Network (S-CRNN), a Convolutional Recurrent Neural Network (CRNN), and an Artificial Neural Network Classifier (ANN) using the DCASE 2017 Task-3 dataset. Relative spectral–perceptual linear prediction (RASTA-PLP) and Mel-frequency cepstrum (MFCC) features are used as input to the proposed multi-model. The performance of monaural and binaural approaches provided to the classifier as an input is compared. In our proposed S-CRNN model, we classified the sound events in the dataset into two sub-classes. When compared with the baseline model, our obtained results show that the PLP-based ANN classifier improves the individual error rate (ER) for each sound event, e.g., the error rate (ER) is improved to 0.23 for heavy vehicle events and 0.32 for people walking, and minor gains are shown in other events as compared to the baseline. Our proposed CRNN performs well when compare to the baseline and to our proposed ANN model. Moreover, in cross-validation trials, the results in the evaluation stage demonstrate a significant improvement compared to the best performance of DCASE 2017 Task-3, reducing the ER to 0.11 and increasing the F1-score by 10% in the evaluation dataset. Erosion and dilation were used during post-processing.

**Keywords:** artificial neural network; Mel-frequency cepstrum; multi-model stacked convolutional recurrent neural network; perceptual linear prediction

## 1. Introduction

The main aim of using sound event detection (SED) [1,2] is to recognize and detect sound events in audio signals, that is, onsets and offsets of events in urban and industrial environments. SED has a wide range of applications, including robots and automated driving, in addition to playing a significant part in processing acoustic signals [3], automatic surveillance of acoustics activities [4], and video retrieval using audio signals [5]. In daily life, we encounter common sound events such as birdsong, dogs barking, human conversation, music, etc. SED implementation faces a number of real-life challenges,

which include intra-class variability, definitive ambiguity, and overlapping sound events. Recognition of such overlapping sound events is referred as polyphonic SED, as shown in Figure 1.

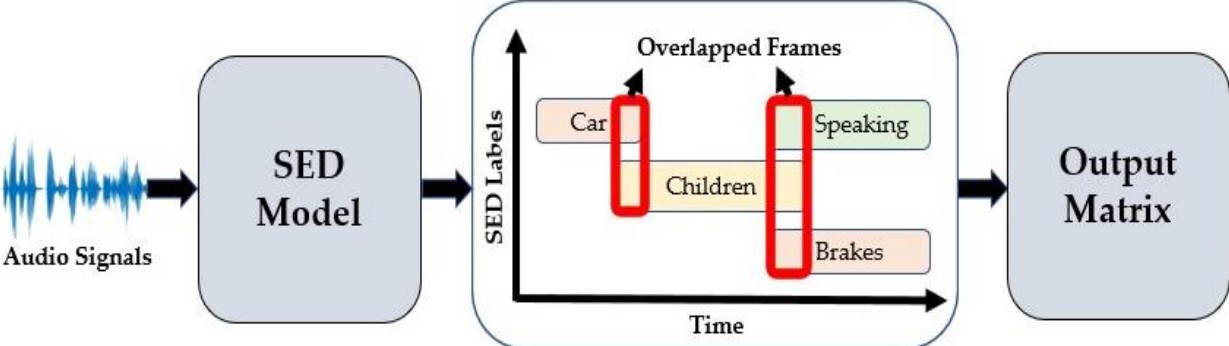

**Figure 1.** A general illustration of SED model.

SED is the process of jointly identifying the various sound event classes and determining the onset and offset times of each individual sound event instance. In Figure 1, where the two sound event classes are active (time) at the same time, this kind of overlapping can be seen. Polyphonic SED is the term used to describe the SED task in such complex sound scenes with overlapping sound events. The most recent polyphonic SED techniques use deep learning-based methodologies. In the order to train an SED model, datasets that include both audio and the related sound event activity annotation are required. This is shown in Figure 1, which illustrates a general outline for developing SED models for an audio dataset that can extract important audio features. Afterwards, a deep learning technique can be employed to transform these features to a sound event as an output matrix.

A number of well-known models [6–10] have been proposed to classify polyphonic SED tasks. For example, Non-negative Matrix Factorization (NMF) [11] can be used as a pre-processing step to create multiple streams of source-separated audio in order to better deal with overlapping sound events. Similarly, the GMM-HMM model [12] can be used to obtain temporal information on overlapping sound events. In addition, multiple DNN-based approaches to address the challenges of polyphonic SED have been proposed to take advantage of the development of deep learning algorithms. An RNN is a type of DNN designed to represent sequence data, including text, voice, and audio data.

The goal of the present study is to create a multi-model that can function effectively with both artificial and natural sounds and to examine how feeding features into the multi-model affects the F1-score and ER. In this article, the available real-life dataset (here, Tampere University (TAU) spatial sound events 2017) is divided into two sub-classes, with the first class consisting entirely of artificial sounds and the second class including natural sounds.

Our proposed multi-model uses two joint features, MFCC and RASTA-PLP, instead of using a single feature for the two sub-classes. In addition, three machine learning algorithms (ANN, CRNN, and S-CRNN) are used with these features for the two sub-classes. Our obtained results show considerable improvements in both F1-score and ER compared to the baseline technique. The glossary/abbreviation section at the end of the article describes the acronyms and abbreviations used throughout in the article. The major contributions of this article are as follows:

- Our proposed stacked multi-model is accurate when used to split and classify artificial and natural sounds, with a high F1-score when fed into the S-CRNN separately.
- For overlapping sounds, that is, polyphonic sound events, CRNN works well compared to alternatives, with the overall ER improved to 0.11 and the F1-score by 10% on the evaluation dataset.

- Finally, for artificial sounds with continuous behavior, such as car sounds, a short filter mask is best; however, for non-continuous sounds, such as speaking, it is preferable to utilize a longer length of filter mask in the post-processing stage.

The rest of the article is organized as follows. Section 2 presents existing works related to the SED model. A brief overview of the DCASE dataset is presented in Section 3. The proposed methodology is described in Section 4. Our results are discussed in Section 5. Finally, Section 6 presents concluding remarks and future research directions.

## 2. Related Work

Using machine learning for DCASE challenges has substantially accelerated SED research in recent years. Various algorithms have been suggested for the DCASE challenge 2013, including HMM, SVM, and GMM, all based on supervised machine learning. The DNN-based classification of polyphonic SED was modelled in DCASE 2015. The DCASE 2016–2022 competitions have triggered a wave of DNN-based SED approaches [13].

The authors in [1] used a single channel log-mel-band energy (MBE) as the audio feature in the baseline technique for the dataset. The network had two FC layers, the dropout and prediction layers, each with 50 units, with the dropout layer having a 0.2 dropout rate. The number of sigmoid units in prediction layer was equal to the number of classes in the dataset. The Adam optimizer and cross-entropy loss were used for training the network over 200 iterations.

In [2], the authors investigated and analyzed the performance of three distinct binaural characteristics, namely, the magnitude and phase components of the short-term Fourier transform, log-mel-band energy, and the extracted log-mel-band energy feature in three different resolution windows. They found that utilizing binaural features produced an error rate comparable to or better than that of single channel features. Using the dataset, it was discovered that the log-mel-band energy feature extracted in various resolution windows produced better best results than the other features.

In [14], the authors evaluated mel-filter bank characteristics with identical bank area and height. They applied a DNN structure to all DCASE 2016 [1] tasks, which performed well on all tasks compared to the baseline [15], except for Task 2. They drew the conclusion that DNNs can be successful in many of these tasks, although they may not be always work well, as in the case of DCASE 2016 Task 2.

In [16], the authors applied Soft-Median Selection (SMS) to smooth out the features of frames. First, a filter called the Differentiable Soft-Median Filter (DSMF) was created for use with neural networks. Second, the SMS was created by combining the DSMF with a Linear Selection (softmax layer). The suggested DSMF addresses the issue of the gradient algorithm failing to smoothly propagate through the median filter.

The authors in [17] used an attention-based capsule network (AttCapsNet) module. They proposed a bidirectional gated recurrent unit (BGRU) module, and a pixel-based attention and bidirectional gated recurrent unit (PBA-AttCapsNet-BGRU) model. Three components of the methods made up the network framework: a CNN audio labelling network, a CNN sound event classification network, and a CRNN feature extraction network.

The study in [18] demonstrates how imposing restrictions during training might help in the design of a CRNN network with a clear functional structure which performs better at SED. For the purpose of locating leaks in the pipes in an industrial setting, the authors suggested a multi-stage Machine Learning (ML) pipeline. Using feature selection approaches, they first minimized the dimensionality of data, then added time correlations by extracting time-based feature which were finally fed to a support vector machine (SVM).

In [19], the authors employed 40 log-mel filter bank coefficients to process monophonic data taken as an input and to normalize the feature output with zero mean and unit variance. They used a bidirectional RNN as their classifier, with 50 hidden units for the input sequence of consecutive 50 frames, and did not apply any post-processing. Performance on the developmental dataset was assessed, and their proposed approach

outperformed the baseline [1]. They came to the conclusion that RNNs are more efficient and adaptable while dealing with varied audio analysis issues.

In [20], a binaural I-Vectors Deep Convolutional Neural Networks (DCNN) and the late fusion method were used as a hybrid approach. For Short-Time Fourier Transform (STFT) post-processing, they used a logarithmic filter bank with 24 bands and 2048 sample windows. They demonstrated that Binaural MFCC shows superior results to the mono-channel approach by comparing Monaural mel-frequency cepstral coefficients MFCC with Binaural MFCC.

In [21], the authors combined an LSTM network with log-mel-band energy characteristics. To extend their model, they built three separate channels and employed several data fusion techniques. Experimental results demonstrated that their methodology outperformed the baseline in terms of performance.

The authors in [22] introduced a CNN model using both short-term and long-term data as input. Additionally, they described a number of optimization techniques, including class-wise early stopping and frequent validation with adaptive thresholds. Comparing the suggested framework to the baseline system, there were noticeable improvements.

To simulate the temporal evolution of sound occurrences, the ref. [23] first introduced a multi label bi-RNN. Additionally, the authors suggested using data augmentation to address the issue of data scarcity and investigated the most effective augmentation techniques to improve performance.

The two channels were combined into one in [24], and the converted spectrum was calculated using the mel-band energy. In this paper, training was carried out utilizing a CNN, showing the feasibility of doing so for SED.

The work in ref. [25] provided a CNN-based SED using a class-wise distance-based approach. By calculating the difference between the audio features of each frame and the class-wise distance of each event, the CNN output could be modified. The detected sound segments were then segmentally re-evaluated using the class-wise distances.

The authors in ref. [26] proposed a more developed version of Neuro-Evolution of Augmenting Topologies (NEAT). They investigated the use of small networks that might compete the much larger networks currently used for SED. In their work, use of k-means clustering and wavelet-based deep scattering transform were employed with NEAT for a more compact representation of the sound input.

The multi-model system put forth in [19] uses DNN to identify car-related sound events. Five models were built on CRNN to identify other sound events, such as children, large vehicles, people speaking, and people walking. Raw audio and log-mel energies were used as input to the features.

In [20], a 20 ms window size was used with a Hamming window and a 10 ms hop-length, and inputs were transformed into a time-frequency representation. MFCC was used as the input feature. The authors divided each audio characteristic into a one-second window, which they then broadcast into a CRNN network. The two primary components of their framework [27] were feature extraction and Bi-LSTM classification.

In the light of above discussion, the existing works in the literature are summarized and compared in Table 1 based on their different SED models and different classification strategies (networks) in different environments with various sound features. In this review, it can be seen that most of the previous works are multi-feature, e.g., existing research works have considered either MFCC, log-mel energy, pitch, or a combination of these. For example, researchers have employed time-domain features, including parameters such as short time energy, zero crossing rate, gradient index, etc., to represent a signal in the time domain, e.g., signal amplitude values. The pitch feature represents the perception of a sound frequency, e.g., low pitch sounds are associated with low frequency sound waves, and high pitch sounds are associated with high frequency sound waves. Similarly, a sound is processed in the frequency domain via frequency filters. It is first converted using Fourier analysis, then multiplied using the filter function, and then transformed back into the spatial domain. The log-mel energy represents frequencies logarithmically (the

corner frequency) over a specific threshold. For instance, in a spectrogram with a linear scale, the vertical space between 1000 and 2000 Hz is half that between 2000 and 4000 Hz. The distance between the ranges is almost the same in the mel spectrogram. However, research work for the SED challenge in urban environments using a combination of MFCC and RASTA-PLP is limited, and requires the attention of engineers and researchers. MFCC is particularly useful for natural sounds. This feature is based on human hearing, which cannot perceive frequencies over 1 Khz. Similarly, the RASTA-PLP feature works better with artificial sounds, which have a greater likelihood of containing noise. The majority of noise-reduction methods used in voice recognition systems heavily rely on applying various filtering strategies of the RASTA-PLP filter across a range of frequency frames; for instance, a band-pass filter can be used to eliminate frames and background noise that changes slowly.

**Table 1.** Summary and comparison of related works on SED models.

| Literature (Year) | SED Model | Classification Strategy/Network | Audio Environment | Feature(s) |
|---|---|---|---|---|
| [1] 2017 | Baseline 2017 | Multi-layer perceptron | Urban | Log-mel energy |
| [2] 2017 | Adavanne TUT 1 | CRNN | Urban | Log-mel energy |
| [8] 2016 | Baseline 2016 | GMM | Urban | MFCC |
| [16] 2022 | Soft-Median Choice | CB+SMC+CDPP | Domestic | Log-mel energy |
| [17] 2022 | Capsule Network | PBA-AttCapsNet-BGRU | Smart cars | Log-mel energy |
| [13] 2022 | Audio Tagging Consistency Constraint | ATCC-CRNN | Domestic | Log-mel energy |
| [18] 2022 | Pipe Leakages | SVM | Pipe leakage | Time-domain features |
| [20] 2017 | Vu-Task3 | RNN | Urban | Log-mel energy |
| [28] 2002 | Viterbi Algorithm | HMM/MLP | Clean Speech | RASTA-PLP Frequency filtering |
| [21] 2017 | Zhou PKU 1 | LSTM | Urban | Log-mel energy |
| [22] 2017 | Lee SNU 3 | CNN | Urban | Log-mel energy |
| [23] 2017 | Lu THU 1 | RNN | Urban | MFCC Pitch |
| [24] 2017 | Chen UR 1 | CNN | Urban | Log-mel energy |
| [25] 2017 | Xia UWA 3 | CNN | Urban | Log-mel energy |
| [26] 2017 | Kroos CVSSP 2 | Neuro-evolution | Urban | Scattering transfer Clustering |
| [19] 2017 | Hu BUPT 2 | BGRU | Urban | Raw audio |
| [27] 2017 | Li Scut 2 | Bi-LSTM | Urban | MFCC |
| Proposed 2022 | Stacked Multi-Model | CRNN, S-CRNN, and ANN | Urban | RASTA-PLP, MFCC |

Therefore, there is a need to develop multi-model classification techniques based on MFCC and RASTA-PLP features for the SED challenge for both natural and artificial sounds in urban environments. The existing schemes in the literature do not address this or use both these features in their SED models. Such an approach could lead to improved overall F1-score and reduced ER. This is the main reason for our proposing a multi-model classification strategy for use in an urban environment.

## 3. Overview of DCASE Dataset

The DCASE dataset used in this article is TUT Sound Events 2017, which is a subset of TUT Acoustic Scenes 2017 [1]. It is made up of acoustic audio recordings that include artificial and natural sounds from different streets with varying amounts of traffic and other activities in urban and suburban environments. Each audio recording is between three and five minutes long, and they are collected in different locations. In daily life, there are six different types of focused sound classes (occurrences) that are quite prevalent. The following sound classes were chosen for the tasks: brakes squeaking, car sounds, children, large vehicles, people speaking, and people walking. Each class was further subdivided into distinct sub-classes such as car engine running, car passing by, and car running; all of these noises are collectively referred to as "car". Similarly, children screaming and children chatting were classed as "children", and "large vehicle" sounds included sounds from buses, trucks, etc. When listening to these audio samples, it can be challenging to distinguish the "person speaking" class of sounds, as voices are quickly masked by background noise from different levels of traffic and other activities. It is very difficult to hear "people walking", as the recordings include other noises that sounds similar. Due to its low sample count and short average length compared to other types of sound occurrences, "brakes squeaking" is difficult to detect in this situation. Neither in the training audio set nor in the test audio set is possible to control how many overlapping sound occurrences occur at each instant. A Roland Edirol R-09 wave recorder with a sampling rate of 44.1 kHz and resolution of 24 bits was used in conjunction with a Binaural Soundman OKM II Klassik/studio A3 electret in-ear microphone to capture the audio events.

Based on the number of examples available for each sound event, the development (Dev-Set) and evaluation datasets (Eval-Set) were separated in the given DCASE dataset. Each recording in the Dev-set was utilized precisely once as test data, and it was made up of four folds, each comprising training and test subsets. Table 2 lists the number of instances for each event subclass in the Dev-Set and Eval-Set.

**Table 2.** Audio event instances per subclass.

| Event (Subclass) | Brakes Squeaking | Car | Children | Large Vehicle | People Speaking | People Walking | Total Occurrences |
|---|---|---|---|---|---|---|---|
| Dev-Set | 52 | 304 | 44 | 61 | 89 | 109 | 659 |
| Eval-Set | 23 | 106 | 15 | 24 | 37 | 42 | 272 |

## 4. Proposed Methodology

In this section, we provide details about our proposed stacked multi-model for SED, as depicted in Figure 2, including sub-sections on the input of audio signals (i.e., events), features extraction, classifier, post-processing, and output. The main aim of this article is to examine the effectiveness of binaural and monaural audio characteristics in order to contrast their outcomes. Below is a detailed description of our proposed methodology used in the neural network for the SED challenge.

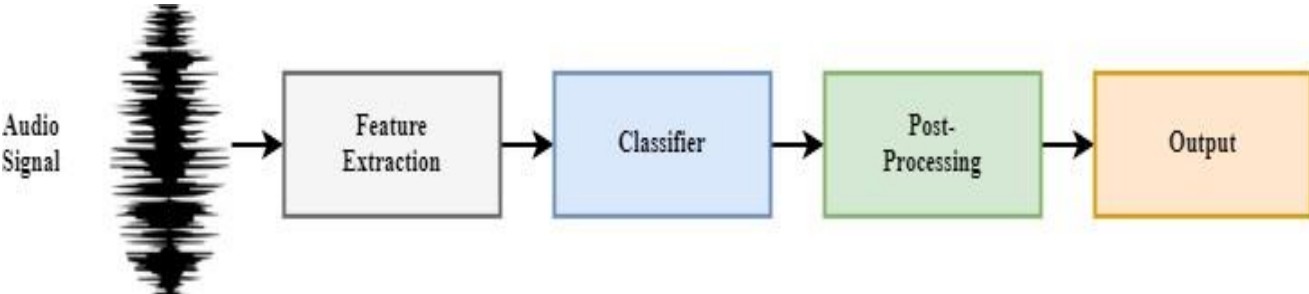

**Figure 2.** Proposed stacked multi-model for SED.

*4.1. Input of Audio Signals*

In this stage, a windowing procedure is applied to the input audio signals in order to divide these continuous audio streams into smaller signal audio segments which are recorded by a microphone. This is accomplished by turning the potentially unlimited signal streams into a continuous series of finite blocks by sliding a window function over the audio stream data.

*4.2. Features Extraction*

In this stage, we use the parameters in Table 3 to find and extract MFCC and RASTA-PLP features in order to effectively extract and differentiate between both binaural and monaural features. Further details are provided in the following sub-sections.

**Table 3.** Parameters used in feature extraction.

| Parameters | MFCC | RASTA-PLP |
|---|---|---|
| Sampling frequency | 44.1 kHz | 44.1 kHz |
| Hop length | 20 ms | 20 ms |
| Mel-filter length | 40 | Nil |
| Model order | Nil | 12 |
| Window | Hamming | Hamming |
| Window size | 1024 | 1024 |
| Do Rasta | Nil | Yes |

4.2.1. MFCC

The MFCC feature is applied to sub-class A as input to the classifier in the proposed SED model. Because humans can hear sounds at frequencies below 1 kHz, this measure is used as the foundation for MFCC feature extraction. Additionally, two feature types, namely, a pre-emphasis filter and triangular-filter, are used by the MFCC; these are spaced logarithmically above 1000 Hz or linearly below 1000 Hz. On the mel-frequency scale, a subjective pitch is provided to capture key phonetic features of speech (i.e., audio signals). The following steps clearly describe the MFCC used in the proposed SED model:

1. The first step in the MFCC is to apply a pre-emphasis filter to increase the strength of the signal at higher frequencies. To decrease noise during audio capture, the following filtering procedure is carried out. Equation (1) presents the input–output connection in the time domain which provides the framework for this filter:

$$y(n) = x(n) - \alpha x(n - 1) \tag{1}$$

   where $x$ is the input audio signal, $\alpha$ is a constant filter coefficient with a value equal to $0.9 \leq \alpha \leq 1$, $y$ is the output audio signal, and $n$ is the time domain.
2. The audio signal is next split into frames within the range of 20 to 40 ms.

3.　A Hamming window is applied to each frame after the audio signal is segmented into frames, as modelled in Equation (2):

$$y(n) = x(n) \times w(n) \tag{2}$$

where $y(n)$ is the output audio signal and $x(n)$ is the input audio signal convolved with $w(n)$, that is, the hamming window calculated in Equation (3):

$$w(n) = 0.54 - 0.46cos\left(\frac{2\pi n}{N-1}\right) \quad 0 \leq n \leq N-1 \tag{3}$$

where $N$ represents total number of frames and $n$ is the time domain.

4.　Using STFT, the time domain data are converted to frequency domain data for each frame $N$, which procedure is formulated in Equation (4):

$$P = \frac{|STFT(x_i)|^2}{N} \tag{4}$$

where $x_i$ denotes the $i$th frame of signal $x$ such that $i = 0, 1, 2,..., N - 1$ and $P$ is the power spectrum.

5.　A triangular filter is used to calculate the filter bank using Equation (5):

$$m = 2595log_{10}\left(1 + \frac{f}{700}\right) \tag{5}$$

where $m$ is the mel scale used to convert an audio frequency into a frequency range that people can hear and $f$ represents the frequency, which can be computed using Equation (6):

$$f = 700\left(10^{m/2595} - 1\right) \tag{6}$$

6.　The inverse logarithm of this log spectrum should be determined after obtaining the relative auditory spectra.

### 4.2.2. RASTA-PLP

The majority of noise reduction methods used in voice recognition systems rely on applying various filtering strategies across a range of frequency frames [28]. Similar to the RASTA-PLP feature extraction methodology, several algorithms are based on frames and high-pass or band filtering methods. This method makes use of the RASTA filter. This filter is a band-pass filter used to eliminate frames and background noise that change slowly. The steps listed below can be used to obtain RASTA-PLP:

1.　First, audio signals are divided into frames.
2.　Then, the logarithm of the short-time critical-band spectrum is determined. The transformation of the bark frequency $\Omega$ from the angular frequency $\omega$ is computed as shown in Equation (7):

$$\Omega(\omega) = 6ln\left\{\omega/1200\pi + \left[(\omega/1200\pi)^2 + 1\right]^{0.5}\right\} \tag{7}$$

3.　Next, we take the regression line-based temporal derivative of the aforementioned spectrum.
4.　Then, an IIR system is utilized to perform temporal derivative filtering on the log-critical band.
5.　In order to imitate human hearing, an equal loudness curve is added.
6.　The inverse logarithm of this log spectrum should be determined after obtaining the relative auditory spectra.
7.　After all of the spectral pole models have been calculated, the PLP requirements are satisfied.

*4.3. Classifier*

In this paper, we use a range of DNN techniques, including ANN, CRNN, and S-CRNN, for the classification of sound events in urban environments. As mentioned in the features extraction stage, we compared and examined both the monaural and binaural features.

We used four hidden states in our proposed artificial neural network (ANN). Each training fold recording was split into frames with a length of 0.1 s, which were then further divided into sub-frames of 25 ms; features were extracted from each frame using a hop size of 10 ms. We initially applied a 0.3 s dilation mask to the output values acquired by the ANN. Dilation was used for zeros that were within 0.3 seconds.

In addition, we used a CRNN model to train local shift-invariant patterns from the acoustic data. We used a filter size of 3 × 3, with batch normalization and max pooling applied after every layer. In order to maintain the temporal resolution of the input, max pooling was only carried out on the frequency axis. In order to learn long-term temporal activity patterns, the layers of CNN are further fed to bidirectional gated recurrent units (Bi-GRU). This is followed by layers of time-distributed fully-connected (dense) layers. In order to produce output with multiple labels, the prediction layer contains a sigmoid activation layer, as shown in Figure 3. The Adam optimizer ([29]) was used, with a learning rate of 0.0001 and a binary cross-entropy loss function [30]. Dropout was used to overcome over-fitting and enhance robustness. The TensorFlow [31] library was used as the backend in Keras, which is an open source software package used to construct artificial neural networks and other design architectures.

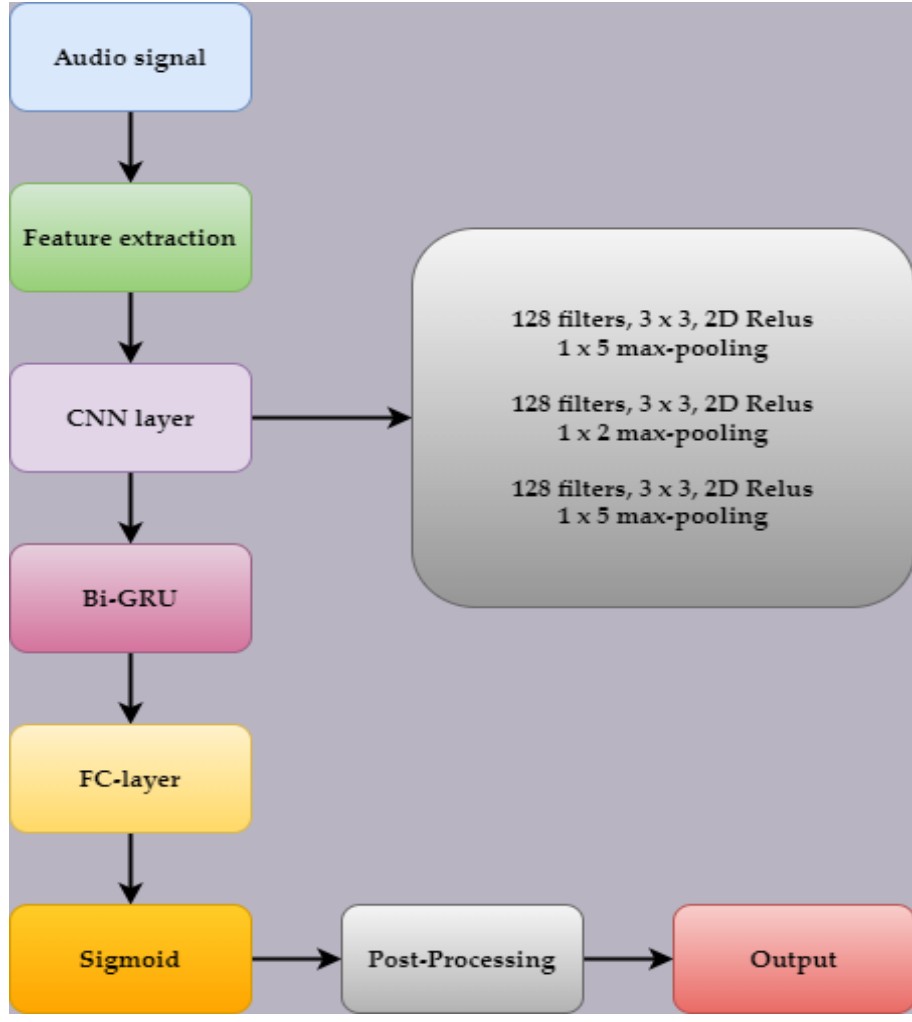

**Figure 3.** Block diagram of CRNN model.

Here, we summarize Figure 3, which represents the working CRNN model. Monaural and binaural features are retrieved from the input audio signal during feature extraction. The extracted features are sent to the CNN layer, which performs a mathematical activity that involves multiplying two functions (sounds) represented as matrices to create a third function, i.e., the output sound. The output is sent to the Gated Recurrent Unit (bi-GRU), which is designed to exploit connections made through a series of nodes in order to carry out machine learning tasks related to memory and clustering, such as speech recognition. The output of the GRU is subsequently sent via a fully connected (FC) layer, which means that a layer is said to be fully connected if all of its inputs are connected to each activation unit of the layer above it. Next, the Sigmoid layer receives the input, on which it employs a sigmoid function to effectively represent a probability. Its range is 0 to 1, and its domain is only real numbers. The sigmoid layer can be used for network layers other than output layers. Erosion and dilation filters are used in the post-processing layer on the output from the sigmoid layer.

Additionally, we employed an S-CRNN which used two concurrent CRNNs with the same set of parameters described above. We refer to this as the S-CRNN model (depicted in Figure 4), as the dataset for the S-CRNN was divided into two subclasses (natural and artificial events), with one subclass fed to one CRNN and the other subclass to the other CRNN.

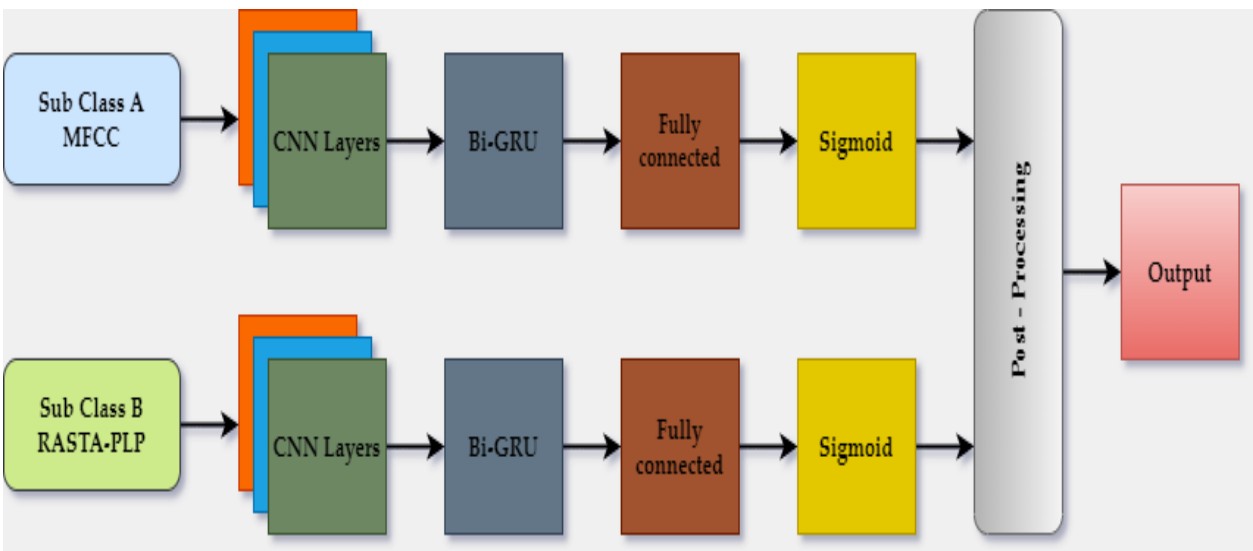

**Figure 4.** Block diagram of S-CRNN model.

In Figure 4, the urban dataset has been split into two subclasses, subclass A (natural sounds) and subclass B (artificial sounds). We used a distinct feature engineering approach for each of the two subclasses. The same CRNN was applied in parallel to the subclasses in distinct sound applications. The CRNNs employed here are the same as those discussed in Figure 3, except two parallel CRNNs are employed instead of one. For artificial sounds with continuous behavior, such as car sounds, a small size was used for the filter mask; for non-continuous sounds, such as human conversation, a larger length of filter mask was used in the post-processing stage. This is the main advantage of parallel CRNN (that is, S-CRNN). RASTA-PLP and MFCC each perform well with both natural and artificial sounds.

### 4.4. Post-Processing

In the post-processing stage, we used several different sizes of erosion and dilation masks. Further, we eroded the post-dilation findings using a 0.2-s erosion mask. Because it is difficult for any class to last for only 0.2 or 0.3 s, we performed both erosion and dilation.

### 4.5. Output

Based on the output from the last recurrent layer, the role of the feed-forward layer is to provide reliable sound event activity prediction. In order to produce output with multiple labels, the prediction layer contains a sigmoid activation layer. The output of the sigmoid layer is defined in Equation (8):

$$\hat{y} = \sigma\left(w^T h + b\right) \tag{8}$$

where $\sigma$ is a sigmoid layer function, $w$ is the weighted vector, $T$ is the transpose, $h$ is the input vector, and $b$ is the bias, which is used to better fit the data.

## 5. Experimental and Performance Evaluation

In this section, we provide the details of the experimental setup using the acoustic sound dataset with the TensorFlow library. All of the implementation programmes were created in an experimental Python environment using Tensorflow for deep learning on a 64-bit desktop with an Intel Core i5-7300HQ quad core processor (6 MB cache, up to 3.5 GHz), 16 GB of 2400 MHz DDR4 memory, 128 GB of SSD storage, 1 TB of 5400RPM SATA storage, and a GTX 1060 6 GB graphics card.

### 5.1. Evolution Metrics

The TensorFlow library was used to implement and analyze the performance of the proposed stacked multi-model for SED detailed in Section 4. TensorFlow is an open-source and free end-to-end software library with Keras as the backend used for multiple machine and deep learning tasks. It provides high-level APIs built in Python for easily training and building deep learning models in deep neural networks. All three models are available in the library, and were modified and trained according to the available DCASE dataset. The evolution metrics used throughout the experiments were the F1-score and ER. The ideal F1-score is 100, which indicates that the system is providing the highest precision and recall values. Conversely, the worst F1-score is 0, which indicates that the system is providing the lowest precision and recall values. The segment-wise F1-score is modelled as formulated below:

$$\text{F1} = \frac{2 \cdot \sum_{k=1}^{K} \text{TP}(k)}{2 \cdot \sum_{k=1}^{K} \text{TP}(k) + \sum_{k=1}^{K} \text{FP}(k) + \sum_{k=1}^{K} \text{FN}(k)} \tag{9}$$

where $k$ presents the number of sound event labels active in both the predictions and ground truth and TP represents true or positive for each segment of one second. The number of sound event labels that are active in predictions and inactive in the ground truth is known as FP, denoting false positive. Similarly, the number of sound event labels that are active in the ground truth and inactive in the predictions is known as FN, meaning false negatives. The second metric, the error rate (ER), is measured using the following Equation (10):

$$\text{ER} = \frac{\sum_{k=1}^{K} S(k) + \sum_{k=1}^{K} D(k) + \sum_{k=1}^{K} I(k)}{\sum_{k=1}^{K} N(k)} \tag{10}$$

where $N(k)$ represents the total number of sound event labels that are currently active in the ground truth of a segment $k$. The following equations are used to calculate the substitutions $S(k)$, deletions $D(k)$, and insertions $I(k)$ for each of the $K$ one-second segments in Equations (11)–(13), respectively:

$$S(k) = min(\text{FN}(k), \text{FP}(k)) \tag{11}$$

$$D(k) = max(0, \text{FN}(k) - \text{FP}(k)) \tag{12}$$

$$I(k) = max(0, \text{FP}(k) - \text{FN}(k)) \tag{13}$$

where $D(k)$ represents the number of reference events that were not correctly identified (false negatives after substitutions $S(k)$ are accounted for) and $I(k)$ represents the number of events in system output that are not correct (false positives after substitutions are accounted for).

*5.2. Results and Discussion*

To evaluate the efficacy of our proposed SED model on accurately solving the SED challenge, we used many feature combinations that we then evaluated and tested using various classifier combinations such as ANN, CRNN, and S-CRNN. After running the TensorFlow experimental model, we obtained the following experimental results, which are tabulated, compared, and discussed in detail below in order to show that the proposed model works as expected.

5.2.1. ANN-Based Classifier

In this section, we compare and analyze the results obtained in the experiment using the ANN classifier on the Dev-Set, which are tabulated in Table 4 below. The baseline model [1] utilizes MFCC for feature extraction, while the ANN-based classifier uses both MFCC and RAST-PLP as feature extraction methods. The results show that applying RASTA-PLP improved both the ER and the F1-Score as compared to the MFCC in the baseline model. For clarity, F-score and F1-score have the same meaning in this article.

**Table 4.** Comparison of results using ANN classifier on Dev-Set.

| Sound Events Class | Baseline [1] | | ANN | | | |
|---|---|---|---|---|---|---|
| | MFCC | | MFCC | | RASTA-PLP | |
| | ER | F1-Score | ER | F1-Score | ER | F1-Score |
| Brakes Squeaking | 0.98 | 4 | 1.00 | 0 | 0.95 | 7 |
| Car | 0.57 | 74 | 0.51 | 55 | 0.47 | 63 |
| Children | 1.35 | 0 | 1.01 | 40 | 1.21 | 0 |
| Large Vehicle | 0.9 | 51 | 0.80 | 27 | 0.67 | 38 |
| People Speaking | 1.25 | 18 | 0.99 | 0 | 0.92 | 10 |
| People Walking | 0.9 | 51 | 0.93 | 10 | 0.58 | 48 |

In Figure 5, the obtained ER results using the ANN-based classifier are compared with the baseline model. The lowest ER was achieved using ANN RASTA-PLP for the sound of squeaking brakes (0.95), followed by those of cars (0.47), heavy vehicles (0.67), people speaking (0.92), and people walking (0.58), while for sound events such as children the lowest ER achieved using ANN with the MFCC feature was 1.01.

Figure 6 depicts the F1-score comparison of the ANN-based classifier with the baseline model. The overall F1-score using RASTA-PLP is not improved; however, when using ANN-based MFCC, the F1-score is improved compared to the baseline model. For example, for large vehicles, when using ANN-based MFCC the F1-score obtained is 27, which lower than the other two extraction features. For car sounds, the baseline F1-score is 74 as compared to other features (i.e., in the range of 70–80).

In summary, when comparing the ER results obtained using the ANN-based classifier to the baseline model, the lowest ER was achieved using ANN RASTA-PLP as a feature input for the sound of squeaking brakes (0.95), followed by those of cars (0.47), heavy vehicles (0.67), and people speaking (0.92), while the lowest ER (1.01) was achieved for people walking (0.58) and sound events such as children by the ANN with MFCC features. As measured by the F1-score, squeaking brakes are detected well using RASTA-PLP, while using MFCC as the input feature works well for detecting sounds made by children as compared to the baseline model.

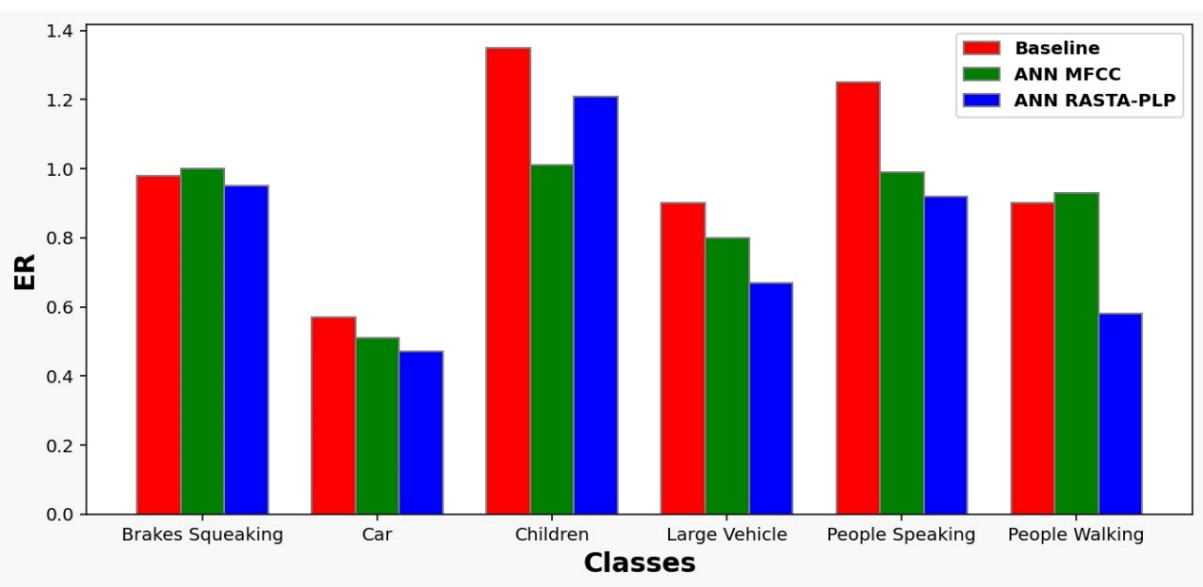

**Figure 5.** ER comparison using ANN classifier.

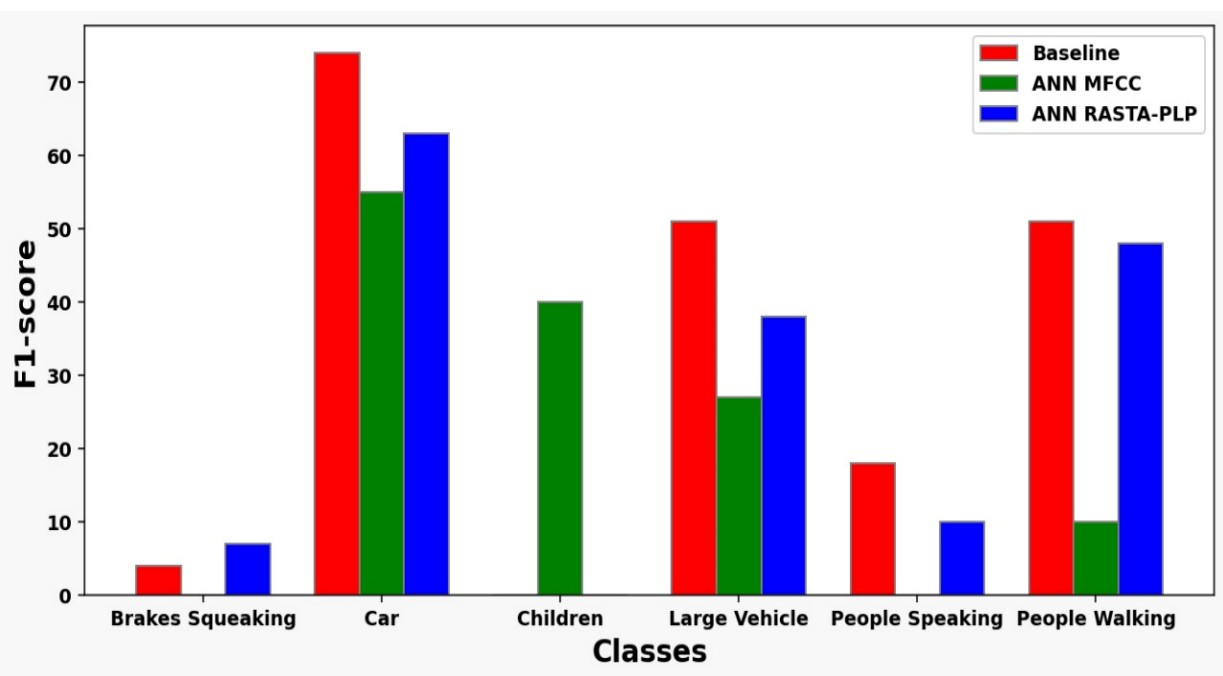

**Figure 6.** F1-score comparison using ANN classifier.

### 5.2.2. CRNN-Based Classifier

This section presents the results obtained in the experiment using the CRNN-based classifier, which employs the same parameters as [2] except that instead of using log-mel energy it uses both monaural features, that is, MFCC, RASTA-PLP, and binaural features, that is, the combination of MFCC and RASTA-PLP. In this experiment, we split the urban dataset into two sub-tasks, namely, machine-generated sounds and natural sounds. For the first task we used a 0.7-s filter mask and the second we used a 0.3-s filter mask. Performance on car and large vehicle sound events was better as filter size is raised in post-processing, whereas for the other sound event classes the performance was worse, and vice versa. The results of the CRNN-based experiments are shown in Table 5.

**Table 5.** Comparison of results using CRNN classifier on Dev-Set.

| Sound Events Class | Baseline [1] | | CRNN | | | | | |
| | MFCC | | MFCC | | RASTA-PLP | | MFCC+ RASTA-PLP | |
| | ER | F1-Score | ER | F1-Score | ER | F1-Score | ER | F1-Score |
|---|---|---|---|---|---|---|---|---|
| Brakes Squeaking | 0.98 | 4 | 0.81 | 0.28 | 0.97 | 0.04 | 0.85 | 0.09 |
| Car | 0.57 | 74 | 0.44 | 0.6 | 0.38 | 0.71 | 0.44 | 0.55 |
| Children | 1.35 | 0 | 0.98 | 0 | 1 | 0 | 1.2 | 0 |
| Large Vehicle | 0.9 | 51 | 0.70 | 0.52 | 0.67 | 0.43 | 0.69 | 0.51 |
| People Speaking | 1.25 | 18 | 0.95 | 0.07 | 0.98 | 0.02 | 0.99 | 0 |
| People Walking | 0.9 | 51 | 0.57 | 0.56 | 0.78 | 0.32 | 0.8 | 0.41 |

Figures 7 and 8 illustrate the performance when using the CRNN-based classifier utilizing hybrid features as input. The comparison shows that the combination of these hybrid features is not as effective as the individual features alone. When comparing individual feature ER and F1-score, MFCC performs better in terms of F1-score, whereas RASTA-PLP performs better in terms of ER.

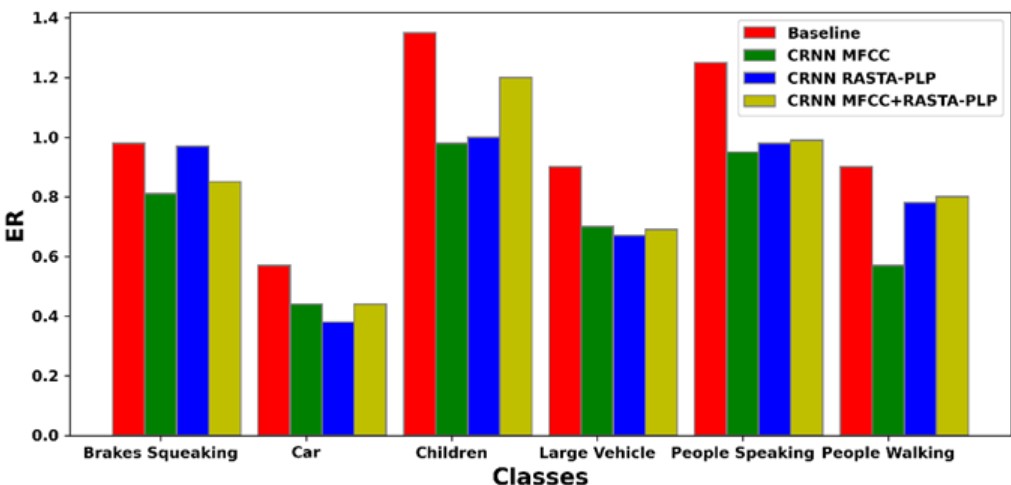

**Figure 7.** ER comparison using CRNN classifier.

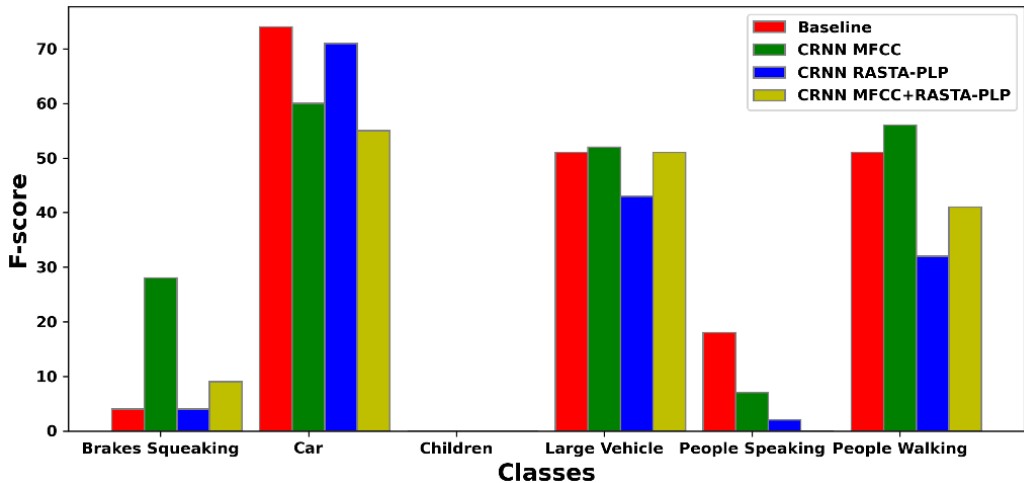

**Figure 8.** F1-score comparison using CRNN classifier.

For example, the MFCC feature in the CRNN for squeaking brakes has the highest F1-score (i.e., 28) and the lowest ER (i.e., 0.81). In the case of car sounds, the RASTA-PLP in CRNN achieves the lowest ER (i.e., 0.38), whereas the baseline [1] has the highest F1-score (i.e., 74). In addition, for children, F1-score achieved is 0, and RASTA-PLP achieves the lowest ER (i.e., 1). In the large vehicle class, the lowest ER is 0.67 and the highest F1-score is 52, both achieved with RASTA-PLP in CRNN. For people speaking, the baseline yields the highest F1-score of 18 and the lowest ER (i.e., 0.95). Similarly, in the people walking class, MFCC in CRNN achieves an F1-score of 56 and an ER of 0.57.

In short, the MFCC feature in the CRNN for squeaking brakes has the highest F1-score (i.e., 28) and the lowest ER (i.e., 0.81). In case of cars, RASTA-PLP in CRNN achieves the lowest ER (i.e., 0.38) whereas the baseline model has the highest F1-score (i.e., 74). In addition, for children, the F1-score achieved is 0, and RASTA-PLP achieves the lowest ER (i.e., 1). In the large vehicle class, the lowest ER is 0.67 and the highest F1-score is 52, both achieved with RASTA-PLP in CRNN. For people speaking, the baseline model yields the highest F1-score of 18 and lowest ER (i.e., 0.95). Similarly, in the people walking class, MFCC in CRNN achieves an F1-score of 56 and an ER of 0.57.

### 5.2.3. S-CRNN-Based Classifier

In this section, we use a multi model classifier that is, S-CRNN, which employs two parallel CRNNs for the two subclasses in the dataset. Subclass 1 includes brakes squeaking, car sounds, and large vehicle sounds, while subclass 2 consists of sound events in the classes of children, people speaking, and people walking as listed in Table 6.

**Table 6.** Comparison of results using S-CRNN classifier on Dev-Set and Eval-Set.

| Sound Events Class | Dev-Set | | | | Eval-Set | | | |
| --- | --- | --- | --- | --- | --- | --- | --- | --- |
| | Baseline [1] | | S-CRNN | | Baseline [1] | | S-CRNN | |
| | MFCC | | MFCC+RASTA-PLP | | MFCC | | MFCC+RASTA-PLP | |
| | ER | F1-Score | ER | F1-Score | ER | F1-Score | ER | F1-Score |
| Brakes Squeaking | 0.98 | 4 | 0.69 | 48 | 0.92 | 16.5 | 0.69 | 55 |
| Car | 0.57 | 74 | 0.33 | 67 | 0.76 | 61.5 | 0.14 | 66 |
| Children | 1.35 | 0 | 0.87 | 17 | 2.6 | 0 | 0.8 | 0 |
| Large Vehicle | 0.9 | 51 | 0.46 | 55 | 1.44 | 42.7 | 0. 9 | 14 |
| People Speaking | 1.25 | 18 | 0.79 | 22 | 1.29 | 8.6 | 1 | 10 |
| People Walking | 0.9 | 51 | 0.48 | 57 | 1.44 | 33.5 | 1 | 54 |

Figures 9 and 10 make it clear that S-CRNN outperforms the baseline model [1] in ER and F1-score across both the Dev-Set and Eval-Set. For example, for car events, the lowest ER of 0.33 is obtained using S-CRNN for the Dev-Set, whereas the baseline [1] achieves an F1-score of 74. The Dev-Set yields the lowest F1-score of 0 for children events and an ER of 1.36 in the baseline model.

Figures 11 and 12 compare the overall ER and F1-score of S-CRNN with the baseline model and state-of-the art performer [2]. Clearly showing that using the S-CRNN model improved the overall F1-score by almost 10 % and the ER to 0.11 on the Eval-Set. In addition, the S-CRNN performed well on the Dev-Set, with an overall improvement of 2% in the F1-score is 0.3 in ER.

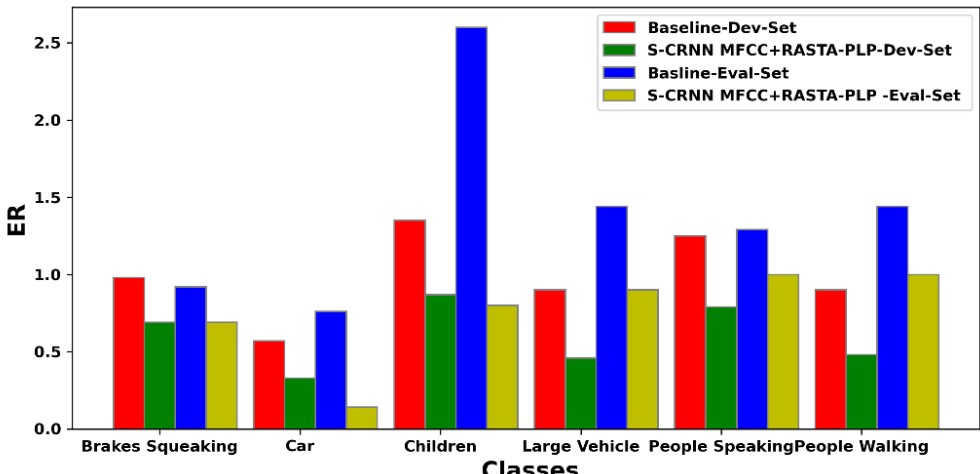

**Figure 9.** ER comparison using S-CRNN Classifier.

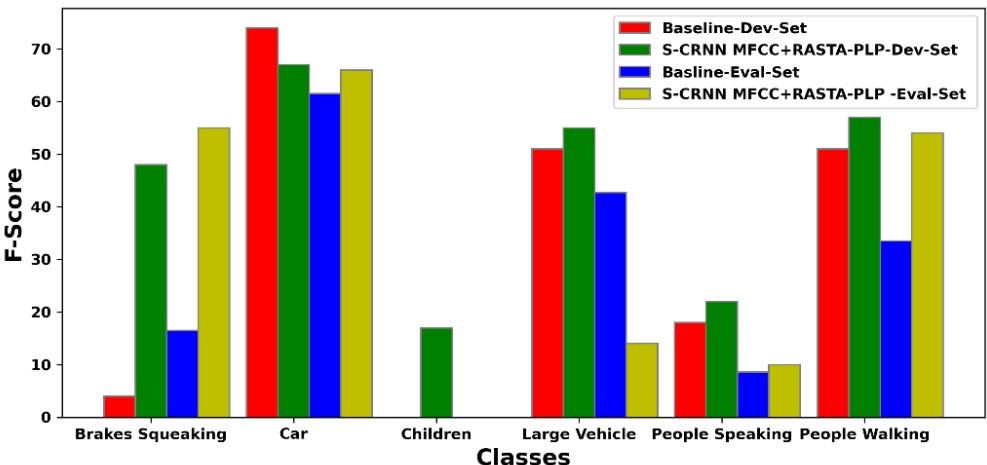

**Figure 10.** F1-score comparison using S-CRNN Classifier.

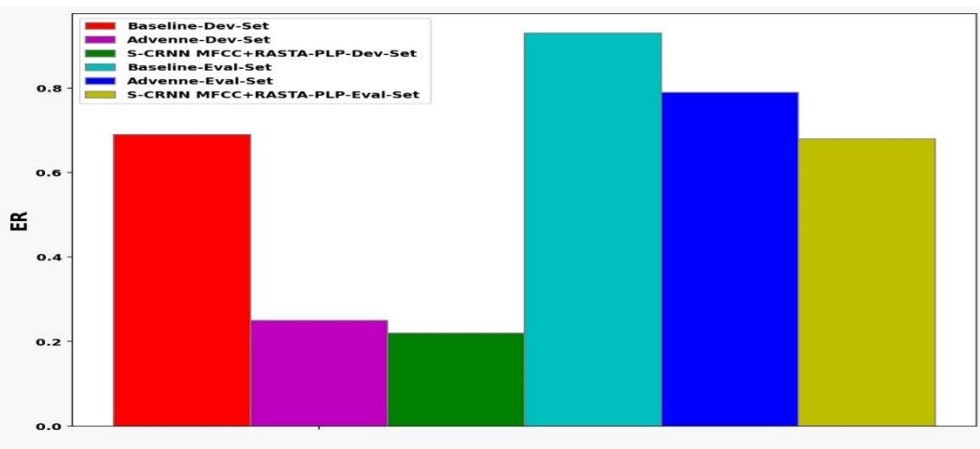

**Figure 11.** Overall ER comparison using S-CRNN Classifier.

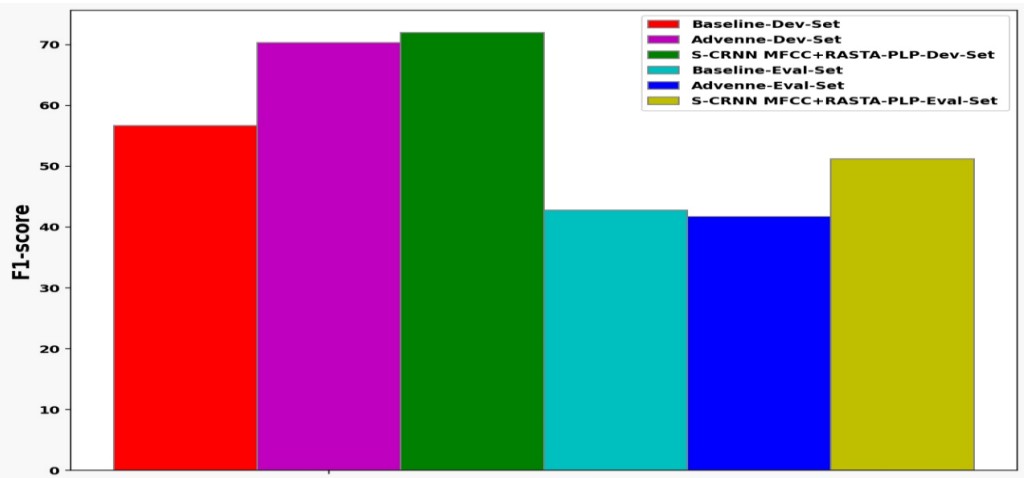

**Figure 12.** Overall F1-score comparison using S-CRNN Classifier.

We evaluated the performance of our proposed S-CRNN model and compared the overall ER and F1-score with the baseline model, as shown in Figures 11 and 12. The baseline model obtains an overall ER of 0.69 and an overall F1-score of 56.7% on the Dev-Set, whereas these scores are 42.8% and 0.93, respectively, on the Eval-Set. On the Dev-Set, the Adavanne model achieves an overall ER of 0.25 and an overall F1-score of 79.3%, whereas on the Eval-Set it achieves an overall ER of 0.79 and an overall F1-score of 41.7%. On the Dev-Set, our proposed S-CRNN model obtains an overall ER of 0.22 and an overall F1-score of 72%, whereas on the Eval- Set the overall ER is 0.68 and the F1-score is 51%. These results clearly demonstrate that the S-CRNN model increased the overall F1-score by over 10% and the ER by 0.11 on the Eval-dataset. Additionally, our proposed S-CRNN performed well on the Dev-Set, with an improved overall ER of 0.3, showing the efficacy of our proposed model.

## 6. Conclusions

In this article, we have introduced a novel multi-model for the SED challenge in urban environments. Because there are several sound event categories which need to be detected and identified in the urban SED task, our main focus in this article was to divide the dataset into two subclasses of audio events prevalent in urban environments. For this, all the machine-generated (i.e., artificial) sound events were placed in the first subclass, while natural sounds such as human conversation were categorized in the second subclass. To achieve this, we employed two distinct features (MFCC and RASTA-PLP spectra) for the two subclasses rather than a single feature for both subclasses. We ran multiple experiments with various ranges of erosion and dilation masks during the post-processing stage. We conclude that continuous noises, such as automobile sounds (excluding brakes), require a modest size of filter mask. As there may be pauses in speech, sounds such as people speaking require a larger size of filter mask. Furthermore, our proposed stacked multi-model (S-CRNN) was able to accurately divide and classify the artificial and natural sound datasets with a high F1-score and low ER. Our future work will focus on developing a lightweight S-CRNN capable to operate well with embedded systems for a number of real-world applications, including robotics, voice enhancement, etc., that need low power and l ow cost. We will try to utilize the real time factor (RTF) in order to determine whether our proposed model will be effective in real-time.

**Author Contributions:** Conceptualization, M.S.K. and M.S.; methodology, M.S.K., M.S. and A.K.; software, M.S.K. and M.S.; validation, M.S., A.K. and M.A.; formal analysis, M.S.K. and L.H.; investigation, M.S. and W.I.; resources, M.S.K. and M.S.; data curation, M.S.K.; writing—original draft preparation, M.S.K., A.A. and E.T.E.; writing—review and editing, M.S.K., M.S., W.I. and A.K.; visualization, M.S.K. and M.S.; supervision, M.S.; project administration, M.S. and L.H.; funding acquisition, A.A. and E.T.E. All authors have read and agreed to the published version of the manuscript.

**Funding:** This research received no external funding.

**Institutional Review Board Statement:** Not applicable.

**Informed Consent Statement:** Not applicable.

**Data Availability Statement:** Not applicable.

**Conflicts of Interest:** The authors declare no conflict of interest.

## Abbreviations

| | |
|---|---|
| ANN | Artificial Neural Network |
| ASR | Automatic Speech Recognition |
| CNN | Convolutional Neural Network |
| DCASE | Detection and Classification of Acoustic Scenes and Events |
| DNN | Deep Neural Network |
| DCNN | Deep Convolutional Neural Network |
| ER | Error rate |
| FC | Fully connected |
| GMM−HMM | Gaussian Mixture Model–Hidden Markov Model |
| GLU | Gated Linear Unit |
| GTCC | Gammatone Cepstral Coefficients |
| GRU | Gated Recurrent Unit |
| IIR | Infinite Impulse Response |
| KNN | k-Nearest Neighbors |
| LSTM | Long short-term memory |
| MFCC | Mel-Frequency Cepstral Coefficients |
| MVDR | Minimum Variance Distortionless Response |
| NMF | Non-negative Matrix Factorization |
| NS | Noise Resistant |
| NB−ACF | Narrow-Band Autocorrelation Function Features |
| NN | Neural Network |
| PLP | Perceptual Linear Prediction |
| RASTA | RelAtive SpecTrA |
| RNN | Recurrent Neural Networks |
| S−CRNN | Stack Convolutional Recurrent Neural Network |
| SED | Sound Event Detection |
| SVM | Support Vector Machine |
| STFT | Short-Time Fourier Transform |
| TF | Time Frequency |
| TAU | Tamper University |
| WPE | Weighted Prediction Error |

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
