# Peer review of "Improved Multi-Model Classification Technique for Sound Event Detection in Urban Environments"

_applsci, doi:10.3390/app12199907_

Round 1

Reviewer 1 Report

The benefit of the method the authors propose is not clear neither from tables nor from figures, they should present the results in a way that reveals the benefit, if there is any. The authors should also discuss possible reasons why, rather than simply presenting the results and pointing out the differences for different types of sounds: for example, why is one method better for one type of sound and another method  for another type of sound?

Author Response

Dear Editor and Reviewer 1

Thank you for the comments concerning our manuscript entitled “Improved Multi Model Classification Technique for Sound Event Detection in Urban Environments” Manuscript ID: applsci-1901693.. Those comments are all valuable and very helpful for revising and improving the quality of our article. We studied the comments carefully and have incorporated the correction to meet the acceptance. Revised portions are marked yellow (highlight) in the manuscript and the responds to the comments are as follow:

Reviewer 1 comments and Author Response:

Comment 01:

The benefit of the method the authors propose is not clear neither from tables nor from figures, they should present the results in a way that reveals the benefit, if there is any.

The authors should also discuss possible reasons why, rather than simply presenting the results and pointing out the differences for different types of sounds: for example, why is one method better for one type of sound and another method  for another type of sound?

Response 01:

We highly appreciate the comment. For possible reason of proposing the multi model classification strategy in this manuscript, please refer to Line 201-215 on page no. 5 Section 2. Short summaries has been added to show the importance and benefits of each method for sound events. Please refer to Lines 486-492 on page no. 13 and 14 for ANN method, Line 524-531 on page no.15 for CRNN method and Line 557-568 on page no. 17 for S-CRNN method. These improvements in results show the efficacy and reasons of the proposed model in this manuscript.

Reviewer 2 Report

The paper is overall well written. I have only a few suggestions. 

I believe that the word "pre-prossessing" (line 66) is misspelled.

There are some minor formatting errors e.g. alignment of equations is non-uniform. 

I would recommend making changes that would prevent the splitting of tables to multiple pages e.g. tables 1 and 3.

Author Response

Dear Reviewer 2

We would like to express our gratitude for the valuable comments and suggestions which have helped us to improve thequality of our manuscript entitled “Improved Multi Model Classification Technique for Sound Event Detection in Urban Environments” Manuscript ID: applsci-1901693. We studied the comments carefully and have incorporated the correction which we hope to meet the acceptance. Revised portions are marked yellow (highlight) .The reviewer comments and detail of the author’s responses with modifications are given below:

Reviewer 2 comments and Authors’ Response:

Comment 01:

I believe that the word "pre-prossessing" (line 66) is misspelled.

Response 01:

We highly appreciate the comment made. We corrected the word, please refer to Line 66 and 67 page no. 2.

Comment 02:

There are some minor formatting errors e.g. alignment of equations is non-uniform. 

Response 02:

We thank the reviewer for this comment. All the equations were aligned.

Comment 03:

I would recommend making changes that would prevent the splitting of tables to multiple pages e.g. tables 1 and 3.

Response 03:

The reviewer comment and suggestion is appreciated. We placed the mentioned Tables (1 & 3) on one page. For further detail, please refer to page no. 5, line 246 in section 2, and page no. 8, line 299 in section 4.2.

Reviewer 3 Report

The authors in the article “Improved Multi Model Classification Technique for Sound Event Detection in Urban Environments” propose and compare multiple neural network models like Stacked Convolution Recurrent Neural Network (S-CRNN), Convolution Recurrent Neural Network (CRNN) and Artificial Neural Network Classifier (ANN) for sound event detection (SED).

The article is structured well and easy to follow. The reader can clearly understand their chosen approach and compare the results with the existing methods. The introduction section 1 gives a quick overview of the problem. Section 2 presents some major and recent works in this area and also provides a summary comparing these approaches. Section 3 describes the DCASE dataset used in this article. The proposed methodology is detailed in section 4 providing an explanation related to filters like MFCC and the pipeline of various processes. Section 5 presents the results followed by a conclusion in section 6.

However, the article in its current form lacks some significant details. Section 2 presents the related works without giving an explanation to some major features like log-mel-energy, pitch etc. We can see them in the features column in Table 1. It is difficult to understand why MFCC and RASTA-PLP features were chosen in this proposed approach. It is also not clear how the authors made the choice of works for this comparison.

In section 4, they present the different models, however, the model summaries are missing. There are some textual descriptions and block diagrams. I would suggest the authors to add the diagrams related to CRNN, S-CRNN and ANN detailing the different layers.

It is not clear how the authors are using F-score. Reading the Evaluation metrics in section 5.1, the reader gets an impression of getting the values between 0 and 1 for F1-score. The authors then quickly talk about F-score (a modified version for handling sound signals?). But it is not clear how F-score is different from F1-score. Later, we see F-score values between 0 and 100.

I would suggest the authors to introduce the necessary terms and equations. Take, for example, the various terms in equations 9 and 10 are not completely explained. They need to discuss the results in detail before the conclusion 6. Figures 11 and 12 are very important and they are not discussed in detail. It is also not clear, whether Tensorflow library contains all the necessary functions for developing these models.

The article needs proofreading. I am highlighting some errors here:

1.     Lines 54 and 55 talk about “frame t” in Figure 1. I do not see any t in the figure.

2.     Line 66 talk about NMF. I would suggest the authors to indicate the readers that there is a glossary in the end of the article.

3.     Line 152: Toextend-> To extend

4.     Table 1: Log mel energy and log-mel-energy are used. Are they different? If not, I would suggest the authors to use a uniform feature name.

5.     Line 212: each comprise of training and test subsets.-> each comprises (or comprising) of training and test subsets.

6.     Line 243: not clear

7.     Figure 6 has some values missing. The reasons are not explained in the text

8.     Line 438: and Figure 12we have compared -> and Figure 12, we have compared

Author Response

Dear Reviewer 3

Thank you for the comments concerning our manuscript entitled “Improved Multi Model Classification Technique for Sound Event Detection in Urban Environments” Manuscript ID: applsci-1901693. These comments are all valuable and very helpful for revising and improving the quality of our article. We studied the comments carefully and have incorporated the correction which we hope to meet the acceptance. Revised portions are marked yellow (highlight) in the manuscript and the responds to the comments are as follow:

Reviewer 3 comments and Authors’ Response:

Comment 01:

However, the article in its current form lacks some significant details. Section 2 presents the related works without giving an explanation to some major features like log-mel-energy, pitch etc. We can see them in the features column in Table 1. It is difficult to understand why MFCC and RASTA-PLP features were chosen in this proposed approach. It is also not clear how the authors made the choice of works for this comparison. 

Response 01:

We carefully read the valuable comments and incorporated the changes in the manuscript. The authors’ response is given below as:

We extended the Related Work in section 2 in order to give significance detail about the mentioned features, literature works as summarized and compared in Table 1. Please refer to page no.4 & 5 and line 185-215 (yellow highlighted) in Section 2.

Comment 02:

In section 4, they present the different models, however, the model summaries are missing. There are some textual descriptions and block diagrams. I would suggest the authors to add the diagrams related to CRNN, S-CRNN and ANN detailing the different layers. 

Response 02:

As per the reviewer recommendation we added the model summaries in this manuscript, please refer to Lines 378-390 page no.10 for CRNN model and for S-CRNN model, please refer to Lines 398-407 on page no. 11 respectively (yellow highlighted).

Comment 03:

It is not clear how the authors are using F-score. Reading the Evaluation metrics in section 5.1, the reader gets an impression of getting the values between 0 and 1 for F1-score. The authors then quickly talk about F-score (a modified version for handling sound signals?). But it is not clear how F-score is different from F1-score. Later, we see F-score values between 0 and 100.

Response 03:

We value the reviewer comment. As the reviewer highlighted we corrected all the miss-spell F-score to F1-score in section 5.1 whose range values are from 0-100, please refer page no. 12, line 435.. Additionally all the F-score words are corrected throughout the manuscript (yellow highlighted). Further, Line 467 clarify both the words on page no. 12.

Comment 04:

I would suggest the authors to introduce the necessary terms and equations. Take, for example, the various terms in equations 9 and 10 are not completely explained. They need to discuss the results in detail before the conclusion 6. Figures 11 and 12 are very important and they are not discussed in detail. It is also not clear, whether Tensorflow library contains all the necessary functions for developing these models.

Response 04:

We appreciate the reviewer commentThe terms in Equation 9 -10 are already described while the terms in Equation 11-13 are described as per reviewer suggestion, please refer to Lines 450-453 page no. 11. Details about Figure 11 and 12 are given and discussed in detail. Please refer to Lines 557-568 page no. 16. Detail about TensorFlow library, please refer to Line 432-433on page no. 11 section 5.1.

Comment 05:

The article needs proofreading. I am highlighting some errors here:

  1. Lines 54 and 55 talk about “frame t” in Figure 1. I do not see any t in the figure.

Response: Correction were made on Line 54 and 55 and “frame t” was removed.

  1. Line 66 talk about NMF. I would suggest the authors to indicate the readers that there is a glossary in the end of the article.

Response: Correction were made on Line 66 and t was removed. Line 83-85 indicate the glossary section to the reader at the end of the article.

  1. Line 152: Toextend-> To extend

Response: Correction were made on Line 153.

  1. Table 1: Log mel energy and log-mel-energy are used. Are they different? If not, I would suggest the authors to use a uniform feature name.

Response: As the reviewer suggested, both have the same meaning, however Log mel energy were changed to log-mel-energy throughout the manuscript for uniformity and are yellow highlighted.

  1. Line 212: each comprise of training and test subsets.-> each comprises (or comprising) of training and test subsets.

Response: As the reviewer suggested, correction was made accordingly at line 273, page no. 7 and are yellow highlighted.

  1. Line 243: not clear

Response: As the reviewer suggested, we made correction for clarity at line 308, page no. 8 and are yellow highlighted

  1. Figure 6 has some values missing. The reasons are not explained in the text

Response: Line 482-483, page no. 13, and figure out the missing value in the Figure 6.

  1. Line 438: and Figure 12we have compared -> and Figure 12, we have compared

Response: We changed it to Figure 11 and 12, we have on line 550 page no. 16.

Round 2

Reviewer 3 Report

Firstly, I thank the authors for considering my review comments in the revised version of their proposed article. The authors have modified section 2 and added a brief introduction of the key sound features that may help the readers understand their contribution. The authors have also justified their multi-modal approach in their use of MFCC and RASTA-PLP.

They have also elaborated the different models used in their proposed approach in section 4. They have clarified the use of F-score and replaced such a term by F1-score. Finally, they have also explained the missing definitions of some terms in the equations 9.

They also discuss the figures 9 and 10 in detail, which will now help the readers to understand their results better.